# What Does CLARITY-BPA Mean for Canadians?

**DOI:** 10.3390/ijerph18137001

**Published:** 2021-06-30

**Authors:** Lindsay D. Rogers

**Affiliations:** Department of Biochemistry and Molecular Biology, University of British Columbia, Vancouver, BC V6T 1Z3, Canada; lindsay.rogers@ubc.ca

**Keywords:** bisphenol A, endocrine-disrupting chemical, xenoestrogen

## Abstract

Bisphenol A is an extremely high-volume chemical widely used in polycarbonate plastics, the linings of food and beverage tins, and shopping receipts. Canadians are ubiquitously exposed to bisphenol A and research shows that exposure at environmentally relevant doses causes endocrine disruption. Recent risk assessments and exposure estimates by the European Food Safety Authority have guided increased restrictions around the use of bisphenol A and established a lower tolerable daily intake, while the CLARITY-BPA program in the United States identified several adverse effects below this exposure level. Within the context of bisphenol toxicity and international regulation, this paper describes the need for revised bisphenol A risk assessments in Canada. Completed in 2008, the most recent bisphenol A risk assessment conducted by Health Canada does not include risks from alternative bisphenols or non-dietary exposure. It also does not account for the additive effects caused by simultaneous exposure to multiple endocrine-disrupting chemicals.

## 1. Introduction

Bisphenol A (BPA) is an extremely high-volume chemical that has been used for over sixty years to produce many items familiar to Canadians (Figure 1A) [1]. Exposure occurs mainly through three types of BPA-containing products. Molecules of BPA are polymerized to form polycarbonate plastic which is used to make products such as water bottles, sunglasses and medical equipment. BPA-derived epoxy resins line the interior of water pipes and most food and beverage tins to prevent corrosion. Most thermal paper used to print documents such as shopping receipts, lottery tickets, and boarding passes is coated with monomers of BPA which functions as a color developer. The global BPA market is increasing by three percent per year and production is projected to reach six million tonnes annually by 2023 [2,3]. Not surprisingly, human exposure is nearly ubiquitous and BPA is detected in more than ninety percent of Canadian urine samples [4]. A tolerable daily intake (TDI) of 25 μg/kg bw/day was established for BPA in Canada in 1996 based on acute toxicity studies in test animals. This value was unchanged during the most recent risk assessment conducted by Health Canada in 2008 [5]. Unfortunately, a now vast and growing number of peer-reviewed studies are reporting adverse effects at much lower doses due to endocrine disruption [6,7,8].

## 2. Regulation of BPA in Canada

Although its estrogenic properties had been previously documented, BPA caused little concern during the first thirty years of widespread use. Then, in 1993, the first data were published demonstrating that BPA can leach from polycarbonate plastic and drive estrogenic responses within cells in culture [12]. Following this work, gestational exposure to doses of BPA that were an order of magnitude lower than the TDI was observed to disrupt prostate development in mice [6]. Numerous studies in test animals expanded upon this finding and concern began to grow around the possibility that low-level exposure to BPA causes reproductive abnormalities, immune and cognitive dysfunction, and metabolic disorders in humans [13,14,15]. In response to this, Canada conducted a Health Risk Assessment of BPA from Food Packaging Applications in 2008 [5]. Formula-fed infants were identified as the highest exposed group where the average probable daily intake (PDI) was estimated at 0.2–0.5 μg/kg bw/day. The assessment identified studies reporting adverse effects below this PDI but the review panel ultimately concluded that “the current dietary exposure to BPA through food packaging uses is not expected to pose a health risk to the general population, including newborns and young children”. Despite this, ALARA (as low as reasonably achievable) was recommended and Canada was the first country in the world to take regulatory action against BPA. In 2008 the Canadian government prohibited the importation and sale of BPA in bottles and food packaging intended for infants and in 2010 BPA was officially declared a toxic compound. While relevant biomonitoring data were not available for Canadian newborns at this time, the above prohibitions were estimated to reduce BPA exposure to bottle-fed infants in Canada by 96%. Unfortunately, infants are not the only relevant group exposed to BPA. Several studies have identified positive correlations between BPA exposure and behavior, cardiovascular disease, and obesity among children [16,17,18]. Data have also demonstrated adverse developmental effects below the PDI for the general population (0.055 μg/kg bw/day) [7,19]. Reduced exposure to children and pregnant women is not anticipated due to the very limited prohibitions implemented in Canada.

## 3. New Data Resolves Discrepancy around BPA Toxicity

For several years considerable discrepancy existed around the estrogenicity of BPA. While studies detected adverse outcomes in test animals exposed to doses as low as 5 ng/kg bw/day, weak estrogenic effects were initially observed in cell culture and in other in vitro experiments [7]. The Canadian health risk assessment completed in 2008 states that “BPA can be estimated to contribute only a minimal quantity of estrogenic activity, in terms of comparison to background 17β-estradiol (E2)”, and the estimated dose of BPA was divided by a BPA/E2 equivalency ratio of 1000. This equivalency ratio is based on the relative binding affinity of E2 and BPA to estrogen receptor α (ERα) and estrogen receptor β (ERβ) in vitro, as well as transactivation assays measuring transcription downstream of estrogen-response elements (EREs) [9,20]. While the estrogenicity of bisphenol A is two to five orders of magnitude lower than E2 as measured by these assays, we now know that BPA can display strong estrogenic responses within several biological systems (Figure 1B). BPA can bind extranuclear estrogen receptors and estrogen-related receptors and induce nuclear responses that are independent of estrogen response elements, all of which can occur within specific cell types at environmentally relevant exposure levels [21,22,23]. For example, a recent study in murine β-cells demonstrated significant ERβ-dependent effects following BPA doses as low as 0.1 nM [8]. The effects did not involve an ERE and thus would not be detected by ERE-transactivation. Bioactivation of BPA to a stronger estrogen in vivo is also a possibility [24]. Due to this, a BPA/E2 estrogenic equivalency ratio of 1000 can no longer be assumed, and exposure to BPA doses as low as 0.025 μg/kg bw/day is likely biologically relevant [19]. 

Beyond its estrogenic effects, BPA has also been shown to bind additional nuclear receptors and alter epigenetic modifications. BPA has displayed anti-androgen activity through in vitro assays in yeast, where the half maximal effective concentration (EC_50_) was 10^−5^ M [25]. BPA also appears to function as a thyroid hormone antagonist, inhibiting transcription mediated by thyroid hormone receptors at concentrations of 10^−6^ M [26]. While the mechanisms behind these effects are largely unknown, a growing number of studies also report epigenetic alterations induced by BPA [27,28]. High-throughput methods such as whole-genome bisulfite sequencing (WGBS), chromatin immunoprecipitation sequencing (ChIP-seq), and assay for transposase-accessible chromatin using sequencing (ATAC-seq), have identified changes in DNA methylation, histone modifications and chromatin accessibility following BPA exposure. Among the numerous changes observed are alterations in microRNA (miRNA). For example, BPA treatment caused significant upregulation of miRNA-146a in two placental cell lines resulting in increased sensitivity to DNA damage [29]. miRNA-146a expression was also induced by BPA exposure in murine testis and was shown to impair steroidogenesis [30]. A recent study in zebrafish suggests that these changes could be transgenerational as hypermethylation of a gene involved in reproductive development was observed across several generations [31]. Exposure to BPA has also been shown to impact gamete quality. Mice exposed to BPA during the fetal-perinatal period showed significantly reduced viability and motility of spermatozoa [32]. At environmentally relevant doses BPA transfers across the human placenta in unconjugated form suggesting that the fetus, which has limited UDP-glucuronosyltransferase activity, is exposed to free BPA [33,34].

## 4. Questioning the Safety of BPA Alternatives

BPA belongs to a larger class of chemicals known as bisphenols which are characterized by two hydroxyphenol functionalities within their structure (Figure 1A). At least fifteen different bisphenols currently exist, many of which can be used to replace BPA in products such as polycarbonate plastic and thermal paper. Bisphenol S (BPS) and bisphenol F (BPF) are currently the most widely used BPA substitutes. Unfortunately, while BPS and PBF production has increased as a corporate response to restrictions and concerns around BPA, current data suggest that they are not safer alternatives [35,36]. In a recent review analyzing thirty-two studies, the estrogenic effects of BPS and BPF were found to be very similar to BPA [37]. Both BPS and BPF display potencies within the same order of magnitude as BPA. Based on relative binding affinity to ERα and ERβ, the average estrogenic potency for BPF compared with BPA was 1.07 and the average estrogenic potency of BPS compared to BPA was 0.32. Interestingly, BPS displays equivalent or greater estrogenic potency to 17β-estradiol when assayed in membrane receptor models [38]. BPS is also the most common BPA replacement used in thermal paper [39]. Numerous adverse effects following low-dose exposure to BPS and BPF in test animals have also been reported. A recent study in the United States detected BPA, BPS and BPF in 96%, 89% and 67% of urine samples collected from Americans [40]. While not used as extensively as BPA and BPF, additional BPA replacements include bisphenol AF (BPAF) and bisphenol B (BPB). Both compounds have been shown to cause endocrine disruption in several species [41,42,43]. A recent study assaying the placental transfer of fifteen bisphenols identified similar rates for BPA, BPF and BPB [44]. Biomonitoring and risk assessment following exposure to BPA replacements does not currently exist in Canada.

## 5. Regulation of BPA Outside of Canada

The European Union has issued scientific opinions on BPA in 2006, 2008, 2010 and 2015. These opinions have provided both dietary and non-dietary exposure estimates, as well as toxicological assessment. Following the scientific opinion issued in 2010, the use of bisphenol A was prohibited in the manufacture of polycarbonate infant feeding bottles, and BPA-containing products are now subject to strict and specific migration limits. Within the opinion issued in 2015, the panel identified significant uncertainty and a need for further research pertaining to the potential reproductive, neurobehavioral, immunological and metabolic effects of BPA [45]. As a result, the panel established a lower temporary TDI of 4 μg/kg bw/day for BPA. Dietary exposure (food and beverage) was identified as the main source of BPA across all population groups. Average dietary exposure among adults was estimated at 0.129 μg/kg bw/day and high exposure was estimated at 0.362 μg/kg bw/day. Average non-dietary exposure (dust, thermal paper, cosmetics) among adults was estimated at 58.9 ng/kg bw/day and high exposure was estimated at 542 ng/kg bw/day. The upper bound for the uncertainty interval for dietary exposure did not exceed the TDI for any age group. However, upper bounds for the uncertainty around high dietary and high non-dietary exposure to BPA were found to exceed the TDI. These wide uncertainty intervals are caused by uncertainty about the magnitude of BPA exposure from thermal paper. The European Union has since prohibited the use of BPA in thermal paper and reduced specific migration limits between two- and twelve-fold for polycarbonate products such as water coolers, kettles, tableware and cookware.

In 2012, the United States launched the Consortium Linking Academic and Regulatory Insights on Toxicity of BPA (CLARITY-BPA) [46]. CLARITY-BPA is supported by three federal agencies, the Food and Drug Administration, the National Institute of Environmental Health Sciences, and the National Toxicology Program. The goal of CLARITY-BPA is to resolve disparities between conclusions drawn by government bodies and academic studies regarding BPA toxicity. Government studies (termed guideline studies) follow validated standardized protocols and typically examine overt signs of toxicity such as organ weight, survival of offspring and histopathology. The designs of academic studies are more varied and aim to identify subtle effects. The CLARITY-BPA study exposed Sprague Dawley rats bred and housed at the National Centre for Toxicology Research to five doses of BPA ranging from 2.5–25,000 μg/kg bw/day. Rats were exposed to BPA from gestational day 6 onwards (continuous dose experiments) or from gestational day 6 through postnatal day 21 (stop-dose experiments). Rats were euthanized at 1 or 2 years of age, and tissues were sent to both government and academic labs for blinded analyses. 

In 2019, results from the guideline studies were made publicly available [47]. At the time of writing, seventeen academic studies have been published [46]. While the guideline studies affirm that current BPA exposure is safe, both the experimental design and interpretation of the data have been challenged. During a 90-day CLARITY-BPA pilot study, BPA and BPA metabolites were detected within the serum of both untreated and vehicle-only gavage treated rats [48,49]. While minimal follow-up experiments were performed, this suggests that at least a population of the rats were exposed to an unidentified environmental source of BPA. In addition, many academic scientists disagree with the gavage method used to administer BPA. Gavage induces stress in the animals, and assessments of brains collected from vehicle-only gavage and untreated animals provided evidence that gavage affected neurodevelopment [50]. 

While the confounding variables outlined above likely diminished differences between control rats and rats dosed with BPA, both the guideline and academic studies identified numerous significant adverse endpoints following BPA exposure (Figure 2A,B). For example, within the guideline studies, increased incidence of mammary adenocarcinoma (females), prostate inflammation (males) and kidney neuropathy (females) were statistically significant endpoints following exposure to 2.5 μg/kg bw/day of BPA (Figure 2A). These results were reported as sporadic by the guideline study authors who dismissed statistically significant results observed following one dose. This has been challenged due to extensive documentation of the nonmonotonic effects of hormones and endocrine-disrupting chemicals [51]. Among the academic studies, the lowest dose of BPA also elucidated the greatest number of effects (Figure 2B). Notably, this dose of 2.5 μg/kg bw/day is 10-fold lower than the current established TDI of BPA in Canada. Applying a 1000-fold safety factor to a LOEL of 2.5 μg/kg bw/day would result in a TDI of 2.5 ng/kg/day, 10000-fold lower than the current value in Canada. Perhaps this is what CLARITY-BPA means for Canadians?

## 6. Canada Needs Revised Assessment around the Toxicity of Bisphenols

In 2008, Canada took the lead in regulatory action around BPA. Since that time the Canadian government has done considerably less than foreign regulatory agencies to update and refine the risks of bisphenol exposure. Canadians need a re-assessment of the literature supporting the endocrine-disrupting properties of BPA. When the most recent risk assessment was completed in Canada, mechanisms rationalizing the low dose effects of BPA in test animals were unclear. We now know that BPA can display strong estrogenicity in various tissues through non-canonical estrogen signaling pathways that are particularly detrimental to developing organisms. The current TDI of BPA in Canada is six-fold higher than the TDI established by the European Food Safety Authority (EFSA), while the results from CLARITY-BPA report adverse effects following doses much lower than the latter. Canadians also need risk assessments for alternative bisphenols, most notably bisphenol S and bisphenol F. Alternative bisphenols are now widely used to replace BPA in polycarbonate, resins and thermal paper. Unfortunately, their physiochemical properties and toxicity appear to be similar to BPA suggesting that products made with these newer chemicals are no safer than the products comprising polycarbonate or BPA-containing resins which were banned more than a decade ago [37]. Finally, Health Canada should initiate non-dietary exposure estimates for bisphenols. Results of the latest scientific opinion on BPA issued by the EFSA concluded that thermal paper is the most abundant source of non-dietary BPA exposure. Most notably, the upper bounds for uncertainty in their high dietary and high non-dietary exposure estimates for BPA exceeded the current TDI. To account for synergistic and additive effects, dietary and non-dietary exposure estimates should be conducted for multiple bisphenols as well as additional endocrine-disrupting chemicals [52,53]. Lastly, in 2017, the House of Commons Standing Committee of Environment and Sustainable Development made eighty-seven recommendations to strengthen the Canadian Environmental Protection Act (CEPA) [54]. Many of these pending amendments will improve the regulation of BPA. These include assessment or reassessment within a prescribed period of time when new data emerges regarding toxicity (recommendation 50), mandatory assessment of cumulative and synergistic effects (recommendation 43), and mandatory assessment of exposures to vulnerable populations, including during critical windows of vulnerability (recommendation 43).

## 7. Conclusions

Bisphenol A is a high-volume chemical that has been used extensively for several decades resulting in nearly ubiquitous exposure to Canadians. In the early 1990s, the estrogenic properties of BPA were realized leading to the restricted use of this chemical in products intended for infants. Since the most recent risk assessment of BPA in Canada, a growing body of peer-reviewed research has characterized molecular mechanisms of BPA toxicity and consistently reported adverse events following low dose exposure in test animals. The EU has reduced the TDI of BPA to 4 μg/kg bw/day and further restricted its use in many products. While any resulting regulatory action is still uncertain, results of the CLARITY-BPA study in the United States indicate that BPA causes adverse effects at doses as low as 2.5 μg/kg bw/day. While factors such as contamination and animal stress likely diminished the differences observed between control and BPA-treated animals in this study, 2.5 μg/kg bw/day is 10-fold lower than the current TDI in Canada, supporting the need to urgently re-assess safe exposure levels for Canadians. In response to restrictions and concerns around BPA, alternatives such as BPS are now widely used but are likely not safer alternatives. Through the Chemicals Management Plan, Health Canada has recently proposed subgrouping 343 BPA structural and functional analogs for future problem formulation [55]. As stated above, this should include a toxicological re-assessment of all bisphenols that Canadians are exposed to. Given the potential for non-monotonicity, all assays should be replicated across a wide dose range and consider various sources, routes and times of exposure. Finally, the synergistic and additive effects resulting from exposure to multiple endocrine-disrupting chemicals must be thoroughly addressed. 

## Figures and Tables

**Figure 1 ijerph-18-07001-f001:**
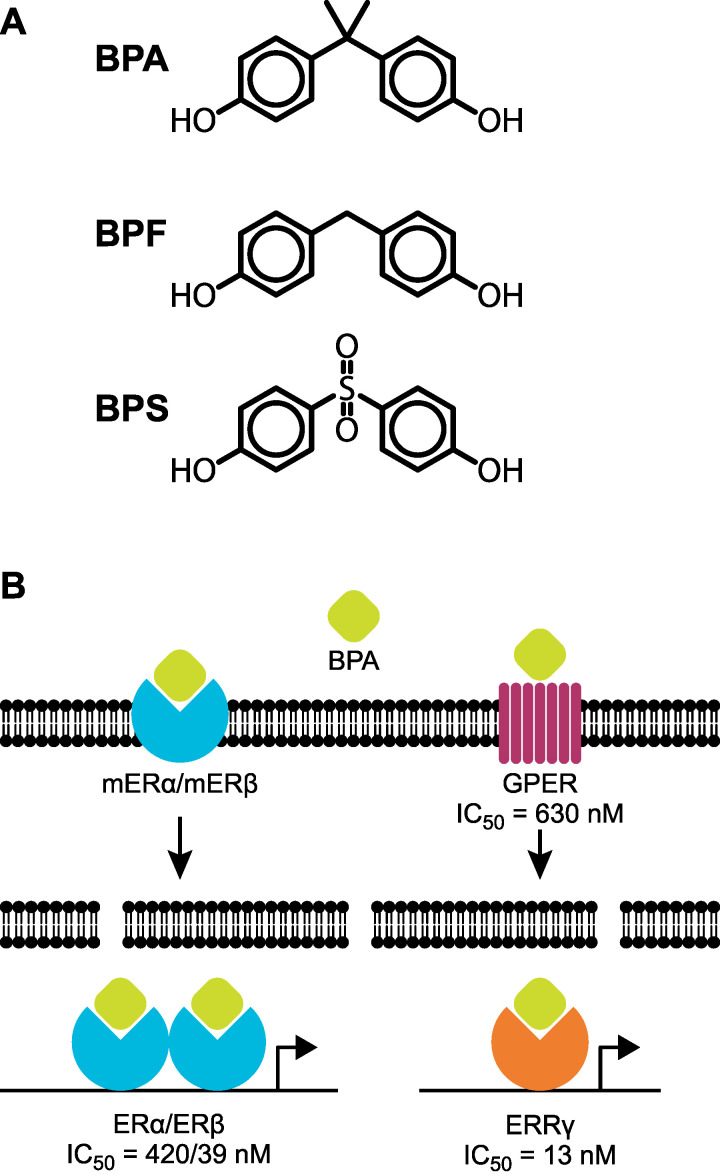
Toxicity of bisphenols. (**A**) Chemical structures of bisphenol A (BPA), bisphenol F (BPF), and bisphenol S (BPS). (**B**) Molecular mechanisms of BPA estrogenicity. BPA can bind nuclear and membrane-bound estrogen receptors α and β (ERα and ERβ), as well as the G protein-coupled estrogen receptor (GPER) and estrogen-related receptor γ (ERRγ). Half maximal inhibitory concentrations (IC_50_) for BPA competing with 17β-estradiol in vitro are listed below each receptor [9,10,11].

**Figure 2 ijerph-18-07001-f002:**
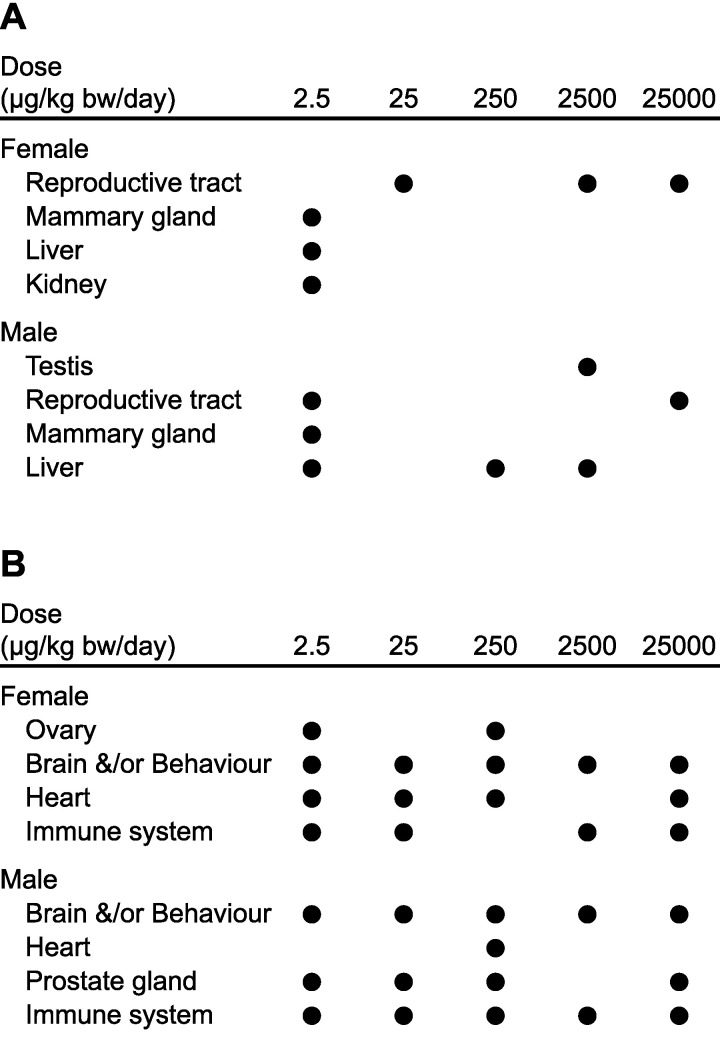
Outcomes from CLARITY-BPA. (**A**) Summary of CLARITY-BPA results from five BPA doses tested within the guideline studies. (**B**) Summary of CLARITY-BPA results from five BPA doses tested within the academic studies. Dots denote significant findings. Modified from Vandenberg LN et al. 2019.

## Data Availability

Not applicable.

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
