# Peer review of "What Does CLARITY-BPA Mean for Canadians?"

_ijerph, 2021, doi:10.3390/ijerph18137001_

Round 1
Reviewer 1 Report
The manuscript entitled: “What does CLARITY-BPA mean for Canadians?” by LD Rogers (Manuscript ID: ijerph-1212512) describes the new bisphenol A (BPA) risk assessments in Canada. The results of the numerous guideline and academic studies imply toxicity and adverse effects following dietary and non-dietary (thermal paper) BPA exposure mostly due to its estrogenic properties.
This is an interesting and well written MS and I have no comments for the author
Author Response
Thank-you for this positive review. The reviewer did not suggest any revisions to the manuscript.
Reviewer 2 Report
Bisphenol A (BPA) is a chemical produced in large quantities for use primarily in the production of polycarbonate plastics and epoxy resins. Human exposure to BPA is widespread.
Article presents new data discrepancy around BPA toxicity and its alternatives. It gives review of Canadian regulation of BPA in the comparison to European Union and United States.
I recommend the paper for publication in International Journal of Environmental Research and Public Health in present form.
Author Response

(The authors gave the same response as above.)

Reviewer 3 Report
The manuscript ijerph-1212512 concerns the BPA toxicity and underlines the need for revised bisphenol A risk assessments. The topic is suitable to the journal. The case of BPA clearly shows that chemicals are not being tested enough, prior to use in mass production. Due to the large number of articles focused on BPA-induced adverse health effects that appear every year, there is a need for an up-to-date review of the literature as well as summary and indication of the direction of further work. The manuscript ijerph-1212512 is a well written perspective on this topic, discussing the most important points and raising the questions to be addressed in the future. Nevertheless, several parts of the manuscript require revision in order to add clarity or to make the manuscript better focused. I have listed some concerns/suggestions below:
- Line 31, 87, 181 – please include literature.
- Other BPA replacements should be also mentioned (like bisphenol B, bisphenol AF).
- It is needed to include the information about BPA transfer to fetus to underline the risk of prenatal exposure.
- Non-monotonic dose response curves, cumulative exposure to mixture of bisphenols, and an additive effect could be also discussed.
Author Response
Thank-you for this positive review. The reviewer suggests four revisions to the manuscript which are addressed below.
- Line 31, 87, 181 – please include literature
Line 31 – Three primary articles are now referenced which demonstrate adverse effects below 25ug/kg bw/day. Due to formatting, these changes now correspond to line 35.
Line 87 – Bisphenol S and bisphenol F market analyses and forecasts are now referenced within the manuscript. Due to formatting, these changes now correspond to line 115.
Line 181 – An article reporting the estrogenicity of BPA substitutes is now referenced within the manuscript. Due to formatting, these changes now correspond to line 220.
- Other BPA replacements should also be mentioned (like bisphenol B, bisphenol AF)
Endocrine-disrupting effects of bisphenol B and bisphenol AF are now referenced and discussed within the manuscript (lines 125-129).
- It is needed to include the information about BPA transfer to fetus to underline the risk of prenatal exposure.
An article demonstrating placental transfer of BPA at environmentally relevant doses is now referenced within the manuscript (lines 104-107). An article comparing materno-fetal transfer rates between 15 bisphenols is also referenced within the manuscript (lines 128-129).
- Non-monotonic dose response curves, cumulative exposure to mixtures of bisphenols, and an additive effect could also be discussed.
Non-monotonicity, synergistic and additive effects are described within the manuscript (lines 15-16, 191-193, 225-227, and 232-233).
Reviewer 4 Report
This manuscript describes very briefly about toxicological studies, risk assessment, and regulations of BPA in the USA and EU, and their implication in risk assessment/management of bisphenols in Canada.
This topic would be relevant for Canadian authority and people; however, the implication of this manuscript to international readers should more clearly be defined by referring to, e.g.,
- How was the TDI 25 μg/kg/day established in Canada
- What is the reason(s)/rational(s) of not updating the TDI by Canadian government by now? What is the reason(s) of not updating the TDI when EFSA determined TDI at 4 μg/kg/day? If the unreliable toxicological finding of the past studies was the reason, please briefly summarize the problems that led to unreliability.
- In what aspects the author regards that the results from CLARITY-BPA project, conducted in the USA, were reliable enough to urge Canadian government considering re-assessment of the BPA risk?
- Please provide briefly more information on dose-effect of BPS and BPF and potential non-dietary (dermal contract?) exposure level to BPA from receipt
- What would be a problem in conducting biomonitoring of bisphenols in Canada for the risk assessment?
etc.
If the implication is adequately defined, then this manuscript would be suitable for publication in IJERPH.
Author Response
Thank-you for this peer review. The reviewer suggests revising the manuscript to increase accessibility by international readers. The reviewer suggests five revisions to the manuscript which are addressed below.
- How was the TDI 25ug/kg/day established in Canada
The first BPA risk assessment was conducted in Canada in 1996. The Bureau of Chemical Safety analyzed data from several reports provided by the Society of the Plastic Industry. A TDI of 25ug/kg bw/day was established based on a 90-day toxicity study in rats which found a no-observable effect level of 25mg/kg bw/day. In 2007 an expert panel re-evaluated BPA toxicity. While the panel acknowledged data demonstrating endocrine disruption at low doses and expressed some concern for adverse effects in pregnant women and fetuses, they considered the data unreliable and inconsistent. The TDI of 25ug/kg bw/day established in 1996 was unchanged by the 2008 health risk assessment. The Health Risk Assessment of Bisphenol A from Food Packaging Applications conducted by Health Canada in 2008 is the most recent health risk assessment in Canada. The manuscript has been revised to include this information (lines 30-33).
- What was the reason(s)/rational(s) of not updating the TDI by Canadian government by now? What is the reason(s) of not updating the TDI when EFSA determined TDI at 4 ug/kg/day? If the unreliable toxicological finding of the past studies was the reason, please briefly summarize the problems that led to unreliability.
As mentioned above, the Health Risk Assessment of Bisphenol A from Food Packaging Applications conducted by Health Canada in 2008 is the most recent health risk assessment in Canada. At the time of this assessment, data concerning endocrine-disruption by BPA were acknowledged but ultimately considered unreliable and inconsistent. At the time of the 2008 risk assessment in Canada, the TDI set by the European Food Safety Authority and the reference dose set by the US Environmental Protection Agency were both 50ug/mg bw/day. The TDI of 4ug/kg bw/day was set by the EFSA in 2015. The US EPA has not updated the reference dose for BPA. The manuscript has been revised to more clearly include this information (lines 30-33).
- In what aspects the author regards that the results from CLARITY-BPA project, conducted in the US, were reliable enough to urge Canadian government considering re-assessment of the BPA-risk?
The goal of CLARITY-BPA was to explain disparities between traditional regulatory studies and findings from independent investigations regarding BPA toxicity. While two significant confounders (BPA contamination and animal stress) undermine the strength of CLARITY data, these variables very likely diminished differences between control animals and animals dosed with BPA. Despite this, a number of adverse effects were observed following exposure to 2.5 ug/kg bw/day within both arms of the study. If 2.5 ug/kg bw/day was accepted as the NOEL for BPA, the TDI in Canada would be reduced by 10,000. Thus, despite limitations due to experimental design, results from CLARITY-BPA support re-assessment of BPA toxicity in Canada. The manuscript has been revised to briefly include this information (lines 245-248).
- Please provide briefly more information on dose-effect of BPA and BPF and potential non-dietary (dermal contact?) exposure level to BPA from receipt.
The manuscript has been revised to include a more thorough description of the dose-effect of BPF and BPS (lines 117-122) as well as placental transfer of these chemicals (lines 128-129).
The manuscript has been revised to include additional details regarding the use of BPA and BPA-replacements in thermal paper (lines 25-27 and 121-122). Additional information regarding non-dietary exposure levels has also been added to the manuscript (lines 147-156).
- What would be a problem in conducting biomonitoring of bisphenols in Canada for the risk assessment?
Biomonitoring of bisphenols is feasible and has been initiated in Canada. In 2007, the Maternal-Infant Research on Environmental Chemicals (MIREC) Study was initiated in Canada which examines the effects of prenatal exposure to environmental chemicals on the health of pregnant women and their infants. Part of the MIREC study includes biomonitoring for bisphenols. However, these data were not available at the time of the most recent health risk assessment for BPA. The manuscript has been revised to clarify this point (lines 55-57).
Reviewer 5 Report
Line 8 – Where is written “recent research shows that exposure at environmentally relevant doses causes endocrine disruption”, I suggest remove the word recent. In fact, PBA and their effects are known for some time;
Line 27 - The authors mention that “BPA is detected in more than ninety percent of Canadian urine samples”. If there are data regarding %s superior to values known to induce effects, mentioning that would enrich the paper.
Line 44 – Similarly, where is written “(PDI) was estimated at 0.2-0.5 μg/kg bw/day”, are there data concerning dose/effects in children?
Line 56 – Please spell out PDI; The same for “ERE” in line 76;
Line 61 – Where is written “in cell culture and in vitro experiments”, it would be more correct to write “in cell culture and other in vitro experiments”;
Line 118 – For all products? If not, which products? One or two examples could be mentioned;
Line 173 – Which “particular times”? Are the authors meaning ages? If so, they could be mentioned;
Line 181 – Please pay careful attention to “than those which were banned more than a decade ago”. Which chemical? BPA? Where? In which products?
The conclusion is not really a conclusion, but a resume. The authors should finish the paper with one or two conclusion sentences.
Line 8 – Where is written “recent research shows that exposure at environmentally relevant doses causes endocrine disruption”, I suggest remove the word recent. In fact, PBA and their effects are known for some time;
Line 27 - The authors mention that “BPA is detected in more than ninety percent of Canadian urine samples”. If there are data regarding %s superior to values known to induce effects, mentioning that would enrich the paper.
Line 44 – Similarly, where is written “(PDI) was estimated at 0.2-0.5 μg/kg bw/day”, are there data concerning dose/effects in children?
Line 56 – Please spell out PDI; The same for “ERE” in line 76;
Line 61 – Where is written “in cell culture and in vitro experiments”, it would be more correct to write “in cell culture and other in vitro experiments”;
Line 118 – For all products? If not, which products? One or two examples could be mentioned;
Line 173 – Which “particular times”? Are the authors meaning ages? If so, they could be mentioned;
Line 181 – Please pay careful attention to “than those which were banned more than a decade ago”. Which chemical? BPA? Where? In which products?
The conclusion is not really a conclusion, but a resume. The authors should finish the paper with one or two conclusion sentences.
Author Response
Thank-you for this positive review. The reviewer suggests 9 minor revisions to the manuscript which are addressed below.
- Line 8 – Where is written “recent research shows that exposure at environmentally relevant doses causes endocrine disruption”, I suggest remove the word recent. In fact, PBA and their effects are known for some time;
Thank-you for this comment. As requested by the reviewer the word recent has been removed (line 8).
- Line 27 - The authors mention that “BPA is detected in more than ninety percent of Canadian urine samples”. If there are data regarding %s superior to values known to induce effects, mentioning that would enrich the paper.
The paper referenced within the manuscript assayed total BPA concentration within Canadian urine samples. The majority of BPA identified within urine samples is the glucuronidated form which has no estrogenic activity. Thus, the aim of the study was to demonstrate that exposure to BPA is continued and widespread. However, a brief discussion regarding materno-fetal transfer of BPA at environmentally relevant doses has been added to the manuscript (lines 104-107).
- Line 44 – Similarly, where is written “(PDI) was estimated at 0.2-0.5 μg/kg bw/day”, are there data concerning dose/effects in children?
Three studies reporting correlations between BPA exposure and behaviour, cardiovascular disease and obesity among children are now referenced within the manuscript (lines 58-60).
- Line 56 – Please spell out PDI; The same for “ERE” in line 76;
The abbreviation probable daily intake (PDI) is defined at the first usage within the text (line 47). The abbreviation estrogen-response element (ERE) is defined at first usage within the text (line 74).
- Line 61 – Where is written “in cell culture and in vitro experiments”, it would be more correct to write “in cell culture and other in vitro experiments”;
The manuscript has been revised to incorporate this suggestion (line 66-68).
- Line 118 – For all products? If not, which products? One or two examples could be mentioned;
Four examples of polycarbonate products are now included, water coolers, kettles, tableware and cookware (lines 156-158).
- Line 173 – Which “particular times”? Are the authors meaning ages? If so, they could be mentioned;
The text “particular times” has been revised to more clearly indicate that endocrine-disrupting effects are especially detrimental following exposure to developing organisms (lines 210-212).
- Line 181 – Please pay careful attention to “than those which were banned more than a decade ago”. Which chemical? BPA? Where? In which products?
The manuscript has been revised to explicitly state that the products banned were particular items comprised of polycarbonate plastic and BPA-containing resins (lines 217-220).
- The conclusion is not really a conclusion, but a resume. The authors should finish the paper with one or two conclusion sentences.
Three concluding sentences have been added to the manuscript (lines 252-256).
Reviewer 6 Report
The large use of BPA as plasticizer represents a possible health risk and regulatory action are still uncertain and debated. The submitted Perspective study analyses BPA regulamentation in Canada vs. USA and Europe stressing the different restrictive approaches in parallel to the main experimental evidence.
In general, the manuscript is well written, but too synthetic, poor in updated references and only focused on BPA toxicology. It does not take into account the emerging epigenetic trangenerational effects described at low BPA doses, the deleterious effects on gamete quality at both high and low doses, and poorly stresses that BPA alternatives are also harmful for health. Furthermore, possible additive effects with other EDCs have to be discussed.
Furthermore, BPA not only interferes in estrogen signalling and this reviewer does not understand the presentation of figure 1B first, and figure 1A later.
Author Response
Thank-you for this peer review. The reviewer makes 6 comments which are addressed below.
- “In general, the manuscript is well written, but too synthetic, poor in updated references and only focused on BPA toxicology”
This is a general comment. The meaning of “too synthetic” is unclear to the author. The manuscript contains many references to primary literature published within the past five years.
- “It does not take into account the emerging epigenetic transgenerational effects described at low BPA doses, the deleterious effects on gamete quality at both high and low doses”
The manuscript has been revised to include a discussion regarding epigenetic effects of bisphenol A as well as transgenerational effects and effects observed in spermatozoa following gestational BPA exposure (lines 90-107).
- The manuscript “poorly stresses that BPA alternatives are also harmful for health”
The manuscript has been revised to discuss BPA replacements more thoroughly. Endocrine-disrupting effects of bisphenol B and bisphenol AF are now referenced within the manuscript (lines 125-128), as well as a comparison of materno-fetal transfer between 15 bisphenols (lines 128-129) and the use of BPA replacements in thermal paper (lines 121-122).
- “possible additive effects with other EDCs have to be addressed”
Possible additive effects among bisphenols and other EDCs are now discussed (lines 15-16, 225-227 and 232-233).
- “BPA no only interferes with estrogen signaling”
The manuscript has been revised to include an additional paragraph describing the effects of BPA on androgen receptors and thyroid hormone receptors (lines 86-90).
- “this reviewer does not understand the presentation of figure 1B first and figure 1A later.
The author is unsure of the meaning of this comment. Figure 1A is presented first within the manuscript (first referenced on line 21). Figure 1B is presented second within the manuscript (first referenced on line 76).
Round 2
Reviewer 4 Report
The authors has revised the manuscript by adding relevant information.
Reviewer 6 Report
The author has addressed most queries and the manucript has been significantly improved. I have not additional query.